# Comparison of the Intestinal Structure and Intestinal Microbiome between Two Geographically Isolated Populations of *Culter alburnus*

**DOI:** 10.3390/ani12030342

**Published:** 2022-01-31

**Authors:** Jun Wang, Bowen Xu, Zhiyi Zhang, Lu Zhou, Guoqi Zhang, Youliang Zhang, Chenghui Wang

**Affiliations:** 1Key Laboratory of Freshwater Aquatic Genetic Resources Certificated by the Ministry of Agriculture and Rural Affairs, National Demonstration Center for Experimental Fisheries Science Education, Shanghai Engineering Research Center of Aquaculture, Shanghai Ocean University, Shanghai 201306, China; wangjun@shou.edu.cn (J.W.); Jefed1997@sina.com (B.X.); zzy19990530@163.com (Z.Z.); 2Shanghai Songjiang Agricultural Development Company Limited, Shanghai 201616, China; luzhou0314@163.com; 3Shanghai Songjiang Aquatic Bred Farm, Shanghai 201616, China; guoqizhang123@sina.cn (G.Z.); youliang1963@126.com (Y.Z.); 4Shanghai Songjiang Aquatic Technology Extension Station, Shanghai 201616, China

**Keywords:** *Culter alburnus*, geographical isolation, intestinal microbiome

## Abstract

**Simple Summary:**

Population differentiation usually forms between geographically isolated populations. We compared the intestinal structures, digestive enzyme activities, and intestinal microbiomes in the Xingkai (XK) Lake and Dianshan (DS) Lake populations of *Culter alburnus*. We sought to discover the differentiated intestinal structure and functional divergence in the two geographically isolated populations. Our study indicated that significantly different intestinal structures, digestive enzymes activities, and intestinal microbiomes were presented in the two populations, which may have been due to adaptative evolution caused by the long-term geographical isolation.

**Abstract:**

Geographical populations of *Culter alburnus* inhabiting different regions of China present substantial differences in their reproduction and development characters. This study compared the intestinal structures, digestive enzyme activities, and intestinal microbiomes in the Xingkai (XK) Lake and the Dianshan (DS) Lake populations of *C. alburnus* collected in two isolated and contrasting river systems. We wanted to discover whether the intestinal structure and functional divergence were formed in the two populations due to adaptive evolution caused by geographical isolation. Our study indicated that higher intestinal villi, thicker intestinal mucosa layer and intestinal muscle layer, and significantly higher activity of α-amylase were identified in the XK population. Moreover, quite different intestinal microbiomes were presented in the two populations, with the higher abundance of Bacteroidetes and Firmicutes in the XK population. The significantly different intestinal microbiome in the XK population was functionally enriched in carbohydrate, lipid, and amino acid metabolism by Kyoto Encyclopedia of Genes and Genomes (KEGG) pathway analysis. Our findings indicated that substantial adaptative divergence in the intestinal structures and intestine microbiomes was formed in the two populations due to long-term geographical isolation, which may have strongly affected the digestion and absorption ability of the XK population compared with the DS population.

## 1. Introduction

Significant population differentiation or environmental adaptation may lead to the formation of new, distinct species, which is a process called speciation [1]. It is well known that population differentiation requires at least partial reproductive isolation, which may result from geographical isolation or isolation by distance (IBD) or other factors [2]. To uncover the basis of the adaptative biological characters of species due to geographical isolation is a research focus in the ecology and evolution scientific community [2,3].

Topmouth culter (*Culter alburnus*) is a fish species with high economic value distributed throughout China [4,5]. Many geographical populations of *C. alburnus* are formed and distributed in Xingkai Lake, Yellow River, Huai River, Yangtze River, and other river systems, which present differentiated population structures and substantial divergent characters [5,6,7]. Among the geographical populations of *C. alburnus* is Xingkai Lake, located in Heilongjiang Province, in the northern part of China (45°20′ N, 132°40′ E). Due to its unique geographic position and water environment, the *C. alburnus* population of Xingkai Lake forms an independent population structure, and presents divergent characters such as the characteristics of eggs, body shape, growth, and development compared to other populations of *C. alburnus* in the southern part of China, such as in the Yangtze River region [5,8,9]. The unique biological characteristics of the Xingkai Lake population make it an excellent model to investigate adaptation and genetic differentiation due to an isolated environment. Although studies have indicated significant population differentiation between the Xingkai Lake and other populations, the underlying mechanism of the adaptative biological characters was not thoroughly investigated in the Xingkai Lake population of *C. alburnus*.

The intestine is an essential organ of animal species, playing roles in food digestion and absorption. The height of the intestinal villi, the thickness of the intestinal mucosal layer and muscle layer, and the related digestive enzyme activities directly regulate the digestion and absorption process of fish species [10,11,12]. Meanwhile, intestine microbiome communities are believed to influence host biology more than previously presumed, and are a crucial factor that defines the hosts’ phenotypes through the so-called “gut-brain axis” [13,14,15,16]. Investigating relationships between intestinal microbiome and their host is essential for a complete understanding of how animals adapt to their specific environment. Whether the Xingkai Lake population of *C. alburnus* has formed an adaptative intestinal structure and intestinal microbiome communities due to isolation has not been fully discovered.

In this study, we conducted experiments on the intestinal structure, digestive enzyme activities, and the composition and function of the intestinal microbiome between the Xingkai Lake and Dianshan Lake populations, which are geographically isolated and contrasting. Our purpose was to uncover whether adaptative divergence has formed in the intestinal structure and intestinal microbiome in the Xingkai Lake population due to geographical isolation.

## 2. Materials and Methods

### 2.1. Animal Sampling

In this study, two *C. alburnus* populations were collected from two geographically isolated lakes, the Xingkai (XK) Lake (Heilongjiang, China) (45°20′ N, 132°40′ E) and the Dianshan (DS) Lake (Shanghai, China) (31°04′ N, 120°54′ E) in 2014 (Figure 1). Both collected XK and DS populations were cultured in the Songjiang Aquatic Seed Farm (Shanghai, China) until they became sexually mature. The artificial breeding of the XK and the DS populations was conducted in the Songjiang Aquatic Seed Farm in May 2020. The offspring of the XK and DS populations were raised in identical artificial ponds on the farm. In December 2020, 10 offspring of the XK and DS populations with bodyweight 25 ± 1.2 g (*p* > 0.05) were collected from two respective ponds. The collected *C. alburnus* individuals (*n* = 10) from the XK and DS populations were immediately dissected. The midgut tissue was sampled, snap-frozen in liquid nitrogen, and stored at −80 °C in a freezer. This study was approved by the Institutional Animal Care and Use Committee of Shanghai Ocean University (Shanghai, China). The sampling procedures complied with the guidelines of the Institutional Animal Care and Use Committee on the care and use of animals for scientific purposes.

### 2.2. Intestinal Structure and Digestive Enzyme Activity Evaluation

The collected midgut tissues from the XK and the DS populations were stored in 4% paraformaldehyde overnight. Then, tissue slices were prepared and stained according to the hematoxylin-eosin (HE) staining method. The slice thickness was set to be 5 um, and after HE staining, the intestinal structure was observed under a microscope system (Leica, Wetzlar, Germany). The height of the intestinal villi, the thickness of the intestinal mucosal layer, and the thickness of the intestinal muscle layer were measured and compared between the two populations [17].

Digestive enzyme activities such as lipase (LPS), α-amylase (α-AMS), and trypsin (Trys) of the midgut tissue were measured in the XK and the DS populations (*n* = 6). The enzyme activities of LPS (U/gprot), the α-AMS (U/gprot), and the Trys (U/gprot) were measured using a lipase assay kit, an α-amylase assay kit, and a trypsin assay kit, respectively (Nanjing Jiancheng Bioengineering Institute; Cat. No. A054-2-1, Cat. No. C016-1-1, and Cat. No. A080-2-2, respectively) and normalized using the total protein content. The data was presented with mean ± SD, and statistical analysis was performed using the Student’s *t*-test.

### 2.3. Sequencing of 16s rRNA and Data Analysis

DNA was extracted from the midguts of the 10 individuals in each population (XK and DS) using the HiPure Stool DNA Kits (Magen, Guangzhou, China) according to the manufacturer’s protocols. The concentration and purity of extracted DNA were evaluated with a NanoDrop 2000 platform (Shanghai, China). The 16S rDNA target region (V3-V4) was amplified by using the 341F (5′-CCTACGGGNGGCWGCAG-3′) and 806R (5′-GGACTACHVGGGTWTCTAAT-3′) primers [18]. The amplified PCR products (~466 bp) were purified using an AxyPrep DNA Gel Extraction Kit (Axygen Biosciences, Union City, CA, USA) and quantified with a QuantiFluor^TM^ fluorometer (Promega, Madison, AL, USA). Then, paired-end sequencing libraries (PE250) were constructed using a TruSeqTM DNA Sample Prep Kit (Illumina, San Diego, CA, USA). The constructed sequencing libraries were sequenced on an Illumina NoveSeq 6000 platform (Illumina, San Diego, CA, USA). The raw reads were deposited into the NCBI SRA database (NCBI accession number: PRJNA792693).

Raw sequencing reads were first quality-filtered to remove the adaptors and the low-quality reads before data processing using FASTP v0.19.6 software [19]. Reads containing more than 10% of unknown nucleotides (N) were trimmed, and reads with less than 50% of bases with quality above 20 were also removed [19]. After filtering, the paired-end reads were merged into a consensus sequence (raw tags) with overlaps longer than 10 bp and mismatch error rates of 2% between sequencing reads by utilizing FLASH v1.2.11 [20]. Noisy sequences of raw tags were further filtered under specific conditions to obtain the high-quality clean tags. Clean tag sequences with ≥97% similarity were clustered into representative operational taxonomic units (OTUs) with UPARSE (9.2.64) [21]. For each individual in the XK and DS populations, the numbers of OTUs were recorded and summarized with USEARCH 7.0 [21]. The representative sequences of each OTU were classified into organisms by a naïve Bayesian model using the Ribosomal Database Project (RDP) classifier with an identity threshold of 0.8 [22]. Comparison of intestinal microbiome communities between groups at the phylum and the genus levels were calculated by Welch’s *t*-test.

The alpha diversity of Sobs, Chao, ACE, Shannon, Simpson, and Pielou’s evenness indices were calculated using QIIME v1.9.1 [23]. Alpha diversity index comparison between groups was calculated by Welch’s *t*-test using the Vegan package v2.5.3 implemented in the R project. The unweighted_unifrac distance matrixes generated by Vegan package v2.5.3 were used to calculate the beta diversity, and were visualized via principal coordinates analysis (PCoA) using the ggplot2 package implemented in the R project. The analysis of similarities (ANOSIM) test was conducted to detect differences between the XK and DS populations using QIIME software with unweighted_unifrac distance [23]. The KEGG pathway analysis of the OTUs was inferred using PICRUSt2 software, and the difference between the two populations was calculated by Welch’s *t*-test [24].

## 3. Results

### 3.1. Comparison of Intestinal Structure and Digestive Enzyme Activity

Histological analysis showed a substantial difference in the intestinal structures between the two populations of *C. alburnus* (Figure 2). The height of intestinal villi (IVH) was significantly higher in the XK than in the DS population (*p* < 0.05), and the thickness of the intestinal mucosa layer (IMT) and intestinal muscle layer (MLT) also were higher in the XK than in the DS population (Figure 2A–C). The enzyme activities of α-AMS, LPS, and Trys also were higher in the XK than in the DS population, and significantly higher α-AMS activity was presented in the XK group (*p* < 0.05) (Figure 2D).

### 3.2. Sequencing and Diversity Estimate of Gut Microbiome between the Two Populations

A total of 2,504,505 paired-end sequences were obtained after sequencing and raw read filtering. After sequence clustering and annotation, a total of 2261 different OTUs, representing 30 bacterial phyla and 482 bacterial genera, were identified (Appendix A). The rarefaction curves of the number of tags sampled and the Sobs index showed clear asymptotes, indicating the nearly saturated sampling of the microbial communities for the two populations (Appendix A).

The XK and DS populations presented different alpha diversity indices, with significantly higher alpha diversity of Sobs, Chao, and Ace indices in the DS group (*p* < 0.05) (Table 1). No significant difference was identified in the alpha diversity of the Shannon and Simpson indices (*p* > 0.05) (Table 1). Beta diversity analysis revealed a clear separation between the XK and DS populations using the PCoA clustering method (Figure 3A). The ANOSIM analysis using unweighted_unifrac distance showed distinct intestine microbiome communities presented in the XK and DS populations (R = 0.586, *p* = 0.001) (Figure 3B).

### 3.3. Comparison of Intestinal Microbiome Compositions between the Two Populations

The XK and the DS populations presented substantially different intestinal microbiome compositions and abundances. At the phylum level, the top five dominant bacterial phyla were Proteobacteria (60.68%), Firmicutes (14.98%), Bacteroides (14.14%), Actinobacteria (4.39%), and Cyanobacteria (1.18%) in the XK population. In contrast, those in the DS population were Proteobacteria (61.98%), Firmicutes (10.88%), Bacteroides (7.89%), Planctomycetes (4.62%), and Verrucomicrobia (4.08%) (Figure 4A). At the genus level, the top three bacterial genera in the XK group were Acinetobacter (23.20%), Bacillus (10.84%), and Flavobacterium (4.70%). In contrast, those in the DS group were Acinetobacter (26.48%), Pseudomonas (4.13%), and Exiguobacterium (3.34%) (Figure 4B).

At the phylum level, the proportions of Bacteroidetes and Armatimonadetes were significantly higher, and Verrucomicrobia was substantially lower in the XK than in the DS population (Welch’s *t*-test, *p* < 0.05; Figure 5A). Meanwhile, at the genus level, a total of 20 genera presented a significant difference between the XK and DS populations (Welch’s *t*-test, *p* < 0.05). The proportion of Cellvibrio, Pseudoxanthomonas, Luteimonas, Devosia, Lysobacter, and other genera were significantly higher in the XK than in the DS population, and only the proportion of Polynucleobacter was significant higher in the DS group (*p* < 0.01) (Figure 5B).

### 3.4. Functional Divergence between the Two Populations

Putative microbiome functions were predicted using PICRUSt2 software. The functional categories, which represented a set of genes influencing the same functional profile, varied significantly between the two populations (Figure 6). In particular, the predicted gene abundance in “lysine degradation”, “valine, leucine, and isoleucine degradation”, “tryptophan metabolism”, and “tyrosine metabolism” categories associated with amino acid metabolism were significantly higher in the XK population (Figure 6; Appendix A). Meanwhile, the predicted gene abundance in the “pentose and glucuronate interconversions” and “ascorbate and aldarate metabolism” categories associated with carbohydrate metabolism and “synthesis and degradation of ketone bodies” category associated with lipid metabolism were also significantly higher in the XK population (Figure 6; Appendix A). In addition, the predicted gene abundance in “Flavonoid biosynthesis” and “Toxoplasmosis” associated with “Biosynthesis of other secondary metabolites” and infectious diseases was higher in the DS population, indicating the significant functional divergence of the two intestinal microbiomes (Figure 6; Appendix A).

## 4. Discussion

Reproductive isolation between or among populations usually causes them to be genetically differentiated, and if maintained for a sufficient period, leads to adaptative evolution and ultimately speciation. The most documented and probably most widespread cause of reproductive isolation is geographic isolation [3]. The XK population of *C. alburnus* has been reported to develop adaptative divergence such as spawning floating eggs, while other geographical populations in the southern part of China, such as the DS population, spawn adhesive eggs [8]. Moreover, the XK population presented different body shapes, karyotypes, and growth characters than the other populations [8,25]. In this study, we identified that the XK population has also formed a different intestinal structure and a divergent intestinal microbiome community, significantly different from the DS population in Shanghai.

The intestinal tract is an essential organ for fish to digest and absorb food resources. The digestion and absorption capacity of organisms can be assessed by the structure of the intestine and the activities of related digestive enzymes. The height of intestinal villi; the thickness of intestinal mucosa and intestinal muscle layers; and the activities of LPS, Trys, and α-AMS directly affect the organisms’ digestion and absorption capacity [11,12]. Our results identified higher LPS, Trys, and α-AMS activities in the XK than in the DS population. Moreover, the height of intestinal villi and the thickness of intestinal mucosa and intestinal muscle layers also showed higher values in the XK population, indicating the strengthened digestion and absorption activity in the XK population. Many studies have pointed out that animals in different temperature regions will form different types of intestinal structures and digestive enzyme activities [26,27]. It was reported that cold acclimation produced significant increases in mean villus height and breadth along the entire intestine in *Cyprinus carpio*, and cold exposure or cold transplantation increased the gut size and absorptive capacity in mice [26,27]. The XK population is located in the northern part of China, which is much colder than the southern part of China. The distinct intestinal structure and digestive enzyme activities of the XK lake may indicate cold adaptative divergence due to the specific environment of the XK lake, which is geographically isolated from the DS lake.

The intestinal microbiome of fish species plays essential roles in digestion, metabolism, and other biological processes [13,28,29]. In our results, both alpha and beta diversity analyses presented significantly diverged intestinal microbiomes between the XK and DS populations. Previous studies indicated that Bacteroidetes could produce carbohydrate metabolism-related enzymes to promote food digestion and provide nutrients to the host [30,31]. Firmicutes promote energy acquisition by improving lipid metabolism [32,33]. The relative abundance of Bacteroidetes and Firmicutes was higher in the XK population than in the DS population. Moreover, the KEGG analysis indicated functional divergence, with higher gene abundance in the carbohydrate, lipid, and amino acid metabolism of the XK population. This finding suggested that the intestinal microbiome may have been shaped into a different community for an altered digestion ability in the XK population compared with the DS population. At the genus level, the abundance of Cellvibrio was also higher in the XK population. Cellvibrio can secrete cellulose-degrading enzymes to promote the digestion of cellulose, hemicellulose, and chitin, which may have benefited the digestion and absorption of cellulose in the XK population [34]. All our results strongly indicated the intestinal microbiome’s composition and functional divergence in the XK lake population compared with the DS lake population may have been due to long-term geographical isolation. The XK and DS lakes are located in the northern and southern part of China, respectively. Other than the significant different water temperature, environmental factors such as the lake area, flow rate, depth of water, and plankton biomass also vary between the two lakes [35,36,37,38]. Although the phytoplankton biomass varies during different seasons in both lakes, bacillariophyta was reported to be the main phytoplankton in the XK lake, whereas chlorophyta is the main phytoplankton in the DS lake [35,37]. All the varied environmental factors may contribute to the distinct intestinal structure and functional divergence of the XK and DS populations.

## 5. Conclusions

In summary, different intestinal structures, digestive enzyme activities, and diverged intestinal microbiomes in the Xingkai Lake population compared with the Dianshan Lake population indicated functionally diverged digestion and absorption abilities, which may have been due to adaptative evolution caused by the long-term geographically isolation in the Xingkai Lake population compared with the Dianshan Lake population.

## Figures and Tables

**Figure 1 animals-12-00342-f001:**
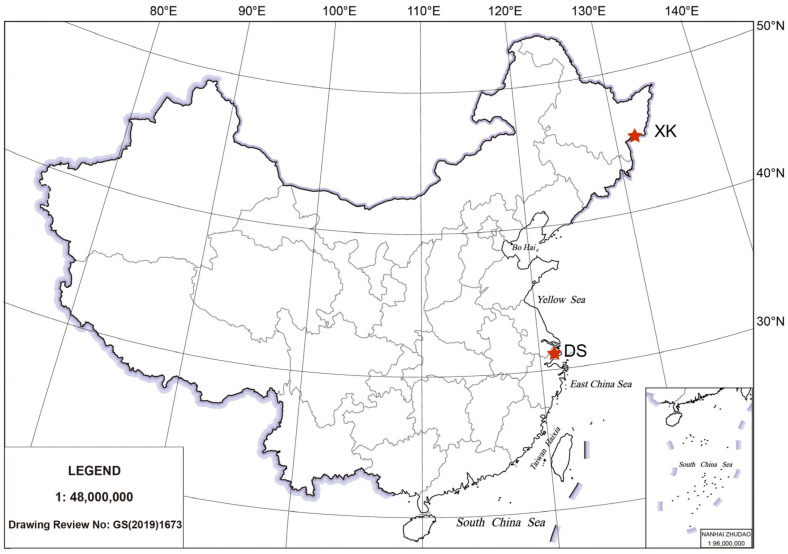
The location of the two geographically isolated lakes: Xingkai (XK) Lake (45°20′ N, 132°40′ E) and Dianshan (DS) Lake (31°04′ N, 120°54′ E).

**Figure 2 animals-12-00342-f002:**
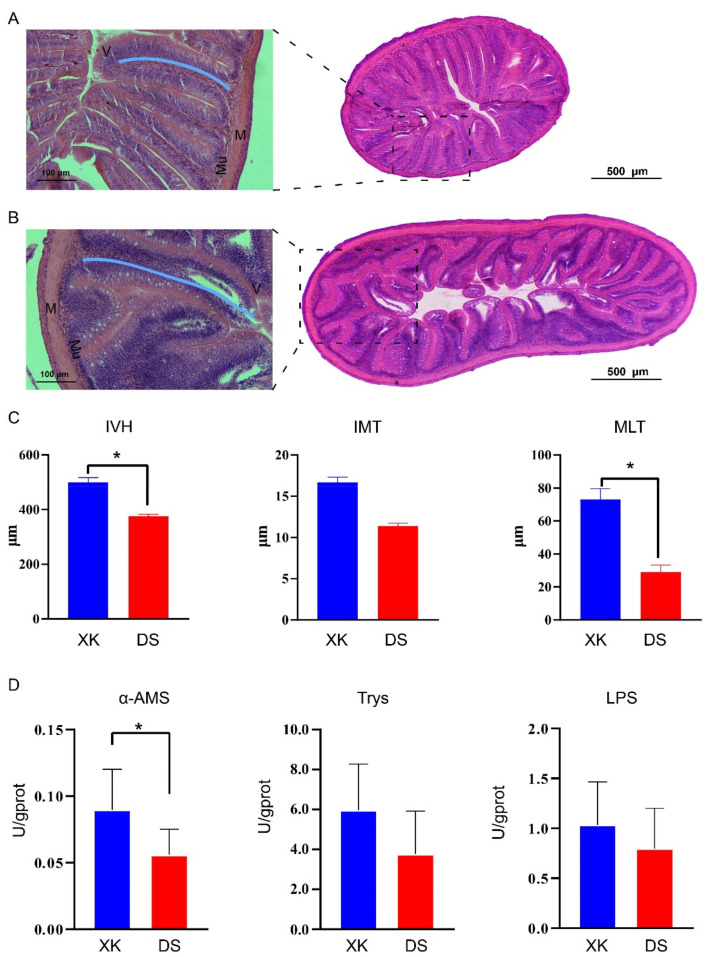
The intestinal structure and related digestive enzymes activities in the Xingkai (XK) Lake and Dianshan (DS) Lake populations of *C. alburnus*. (**A**) Intestinal tissue slices from Dianshan Lake population. (**B**) Intestinal tissue slices from Xingkai Lake population. M, muscle layer; V, intestinal villus; Mu, intestinal mucosa. The blue line indicates the intestinal villus height. (**C**) Measurements of the intestinal structure for the Xingkai (XK) Lake and Dianshan (DS) Lake populations. IVH, intestinal villi height; IMT, intestinal mucosa thickness; MLT, intestinal muscle layer thickness. * indicates *p* < 0.05. (**D**) Digestive enzyme activity (α-amylase, trypsin, and lipase) between the Xingkai (XK) and Dianshan (DS) populations (*n* = 6). * indicates *p* < 0.05.

**Figure 3 animals-12-00342-f003:**
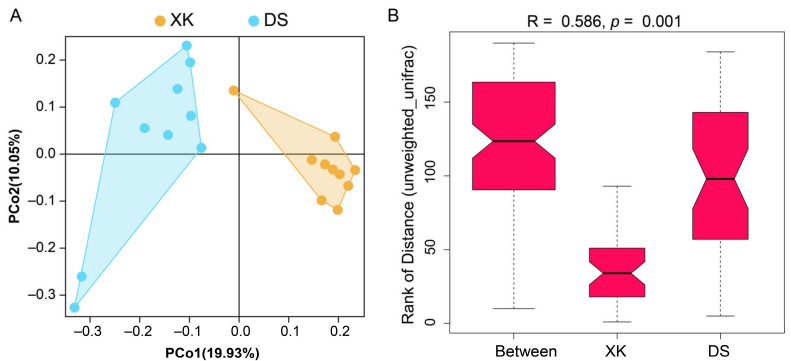
Beta diversity analyses of the intestinal microbiota between the Xingkai (XK) Lake and Dianshan (DS) Lake populations of *C. alburnus*. (**A**) PCoA estimates for the intestinal microbiome communities of the XK and DS populations. (**B**) ANOSIM estimate for the XK and DS populations.

**Figure 4 animals-12-00342-f004:**
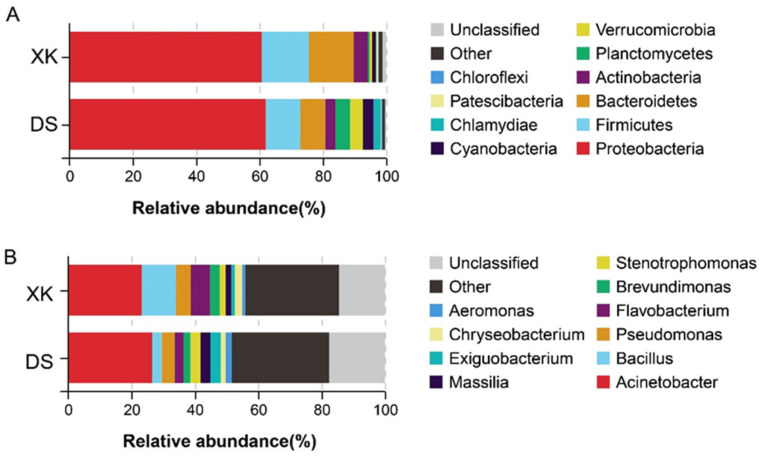
Composition and proportion of the intestine microbiome of the Xingkai (XK) Lake and Dianshan (DS) Lake populations of *C. alburnus* at the phylum level (**A**) and the genus level (**B**).

**Figure 5 animals-12-00342-f005:**
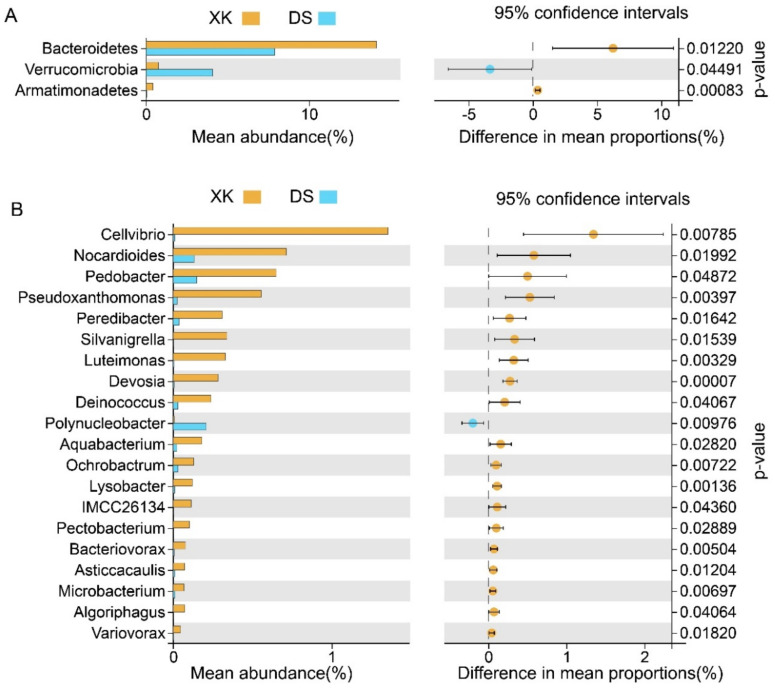
Comparison of the intestine microbiome communities between the Xingkai (XK) Lake and the Dianshan (DS) Lake populations of *C. alburnus* at the phylum level (**A**) and the genus level (**B**).

**Figure 6 animals-12-00342-f006:**
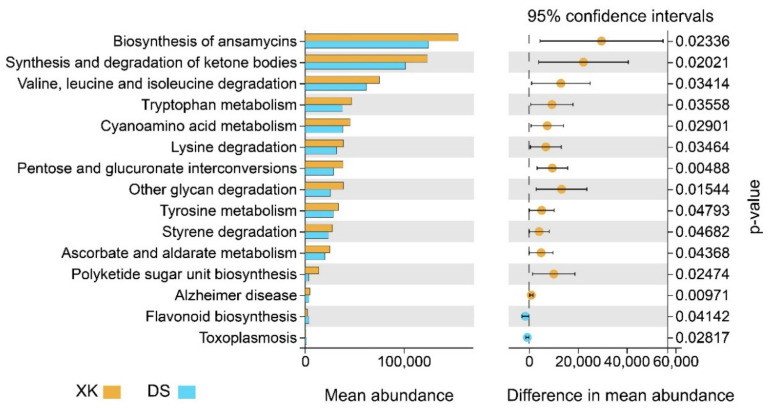
Functional prediction of the intestine microbiome communities in the Xingkai (XK) Lake and the Dianshan (DS) Lake populations of *C. alburnus*.

**Table 1 animals-12-00342-t001:** The alpha diversity indices of intestinal microbiota in the Xingkai (XK) Lake and Dianshan (DS) Lake populations.

Group	Sobs	Chao	Ace	Shannon	Simpson	Pielou
XK	407 ± 55.77 ^b^	520.93 ± 81.74 ^b^	529.16 ± 82.3 ^b^	5.46 ± 1.05 ^a^	0.9 ± 0.11 ^a^	0.63 ± 0.12 ^a^
DS	566 ± 194.14 ^a^	691.35 ± 174.83 ^a^	713.43 ± 166.88 ^a^	5.25 ± 1.21 ^a^	0.89 ± 0.07 ^a^	0.58 ± 0.12 ^a^

Sobs (observed species) index indicates the number of identified OTUs; Chao and Ace indices indicate the number of predicted OTUs. The Sobs, Chao, and Ace indices reflect the species richness level. The Shannon and Simpson indices reflect the species richness and evenness level overall. Pielou indicates Shannon’s evenness. Different letters indicate significant statistical differences (*p* < 0.05).

## Data Availability

All raw read sequences have been uploaded to the NCBI sequences read archive under BioProject accession number PRJNA792693.

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
