# Peer review of "Comparison of the Intestinal Structure and Intestinal Microbiome between Two Geographically Isolated Populations of Culter alburnus"

_animals, 2022, doi:10.3390/ani12030342_

Round 1

Reviewer 1 Report

Geographically isolation usually leads to adaptative evolution and makes isolated populations form distinct biological characters, which will affect the growth, development, immunity, reproduction, and so on. The manuscript “Comparison of the intestinal structure and intestinal microbiota between two geographically isolated populations of Culter alburnus” investigated the adaptative divergence of the intestinal structure and microbiota between two isolated C. alburnus population. The study was very interesting, the methods and analysis were reasonable, and the results were solid. I have a few comments as follows. 

  • As you mentioned there are many populations of C. alburnus in China, why did you only choose XK and DS to do the research? Meanwhile, you mentioned the special biological characters of the XK populations, such as spawning floating eggs. Do any other populations also have this character?
  • If the two populations have such different biological characters (floating eggs vs adhesive eggs), are they two species or two populations?
  • More details should be provided for the materials and methods part. You mentioned 10 individuals were sampled, however, in Figure 1C, you indicated n =6 for the enzyme activity detection. Which tissue was used for digestive enzyme activity detection? Midgut or blood?
  • How did you measure and calculate intestinal mucosa thickness? The figure 1 labeled muscle layer and intestinal villus but not intestinal mucosa.
  • The reference format was not consistent, some presented the full name of the journal, some presented abbreviations. The authors should follow the journal’s requirement.

Author Response

Response to reviewer 1 comments

Comments and Suggestions for Authors

Geographically isolation usually leads to adaptative evolution and makes isolated populations form distinct biological characters, which will affect the growth, development, immunity, reproduction, and so on. The manuscript “Comparison of the intestinal structure and intestinal microbiota between two geographically isolated populations of Culter alburnus” investigated the adaptative divergence of the intestinal structure and microbiota between two isolated C. alburnus population. The study was very interesting, the methods and analysis were reasonable, and the results were solid. I have a few comments as follows. 

As you mentioned there are many populations of C. alburnus in China, why did you only choose XK and DS to do the research? Meanwhile, you mentioned the special biological characters of the XK populations, such as spawning floating eggs. Do any other populations also have this character?

Response: Thank you for your comments. XK lake locates in the northern part and DS lake locates in the southern part of China. The XK population of C.alburnus is believed to be the most northern part population and has formed substantial different characters compared to southern population such as the DS lake population. During the culture experiment in Shanghai, we also discovered that the two populations have different reproduction and growth characters, in order to compare the biological difference between them, we selected the two populations to conduct the experiment. We are aware that other populations of C.alburnus may have different biological characters but they are not our focus in this study. As far as we know, XK population of C.alburnus is the population that has been reported to spawn floating eggs among all the populations of C.alburnus.  

If the two populations have such different biological characters (floating eggs vs adhesive eggs), are they two species or two populations?

Response:Our karyotype experiment indicated there are 2n = 48 chromosomes in both populations and artificial hybridization experiments between the XK and DS populations generated live offspring (unpublished data). Meanwhile, population genetics research also indicated the XK is a geographical population of C.alburnus [1,2]. We think the XK is a population of C.alburnus with significant genetic differentiation compared to the DS population. However, more research is required to investigate the ‘speciation’ issue of XK population in future study.  

[1] Sun, N.; Zhu, D.-M.; Li, Q.; Wang, G.-Y.; Chen, J.; Zheng, F.; Li, P.; Sun, Y.-H. Genetic diversity analysis of topmouth culter (Culter alburnus) based on microsatellites and d-loop sequences. Environ. Biol. Fishes 2021, 104, 213-228.

[2] Qi, P.; Qin, J.; Xie, C. Determination of genetic diversity of wild and cultured topmouth culter (Culter alburnus) inhabiting china using mitochondrial DNA and microsatellites. Biochem. Syst. Ecol. 2015, 61, 232-239.

More details should be provided for the materials and methods part. You mentioned 10 individuals were sampled, however, in Figure 1C, you indicated n =6 for the enzyme activity detection. Which tissue was used for digestive enzyme activity detection? Midgut or blood?

Response: We sampled 10 individuals for the 16s rRNA sequencing and 6 individuals for the digestive enzyme activities detection. Midgut tissues were used for the digestive enzyme activity detection.

How did you measure and calculate intestinal mucosa thickness? The figure 1 labeled muscle layer and intestinal villus but not intestinal mucosa.

Response: We have labeled the intestinal mucosa (Mu) in the revised figure.

The reference format was not consistent, some presented the full name of the journal, some presented abbreviations. The authors should follow the journal’s requirement.

Response: Thank you for the comments, we have revised the format of the references.

Reviewer 2 Report

SUMMARY

The aim of this study is determining if there are differences in the intestinal microbiota, enzyme activity and structure of the intestine of two isolated populations of Culter alburnus from different geographical locations. And if those differences may be due from geographical isolation.

This manuscript contributes to the knowledge whether evolutive adaptation could significantly modify gut structure and enzymatic activity as well as microbiota of animals. This study shows that there are differences in microbiome composition and diversity and also in the physiology ad enzyme activity of gut. Finally, microbiome functionality analysis was done, in order to show if those microbiome differences are related with an adaptative response.

GENERAL CONCEPT COMMENTS

The word microbiota in the title should be changed to microbiome, which is more adequate to metagenomics analysis in which only 16S rRNA gene is used. Microbiota refers bacterial gut composition and microbiome to the OTUs composition.

In general, methodology procedures are well explicated. Only a few comments.

In the text is not mentioned how many individuals of each population (XK and DS) are sampled for this study. And if animals for each population are sampled from the same pound or from different pounds (L86). This point is important, because in all experiments is necessary have at least three biological duplicates of each treatment; in this case of each population.

Another study that could be interesting to done, maybe another time, is sampling wild individuals at different stations. This could give more information about gut parameters differences of each population due to their speciation to live in its environment. In the manuscript here presented,  in the results obtained of the offspring after being cultured during 6 years in controlled environment, is likely some species are loosed and maybe microbiome diversity has been modified, as well as intestine enzyme activity and structure. The sentence of L283-284 must be supported by some references.

This study has an interesting objective, that is dilucidated microbiome implication in fish adaptation to specific ecosystems, by comparing two different populations spatially differentiated. But, authors only apport temperature information about these two ecosystems. It may be attractive to know a bit more about other parameters that differentiate both lakes, and if those could explain the differences herein obtained in these two populations.

Finally, in my opinion, Discussion should be a bit more extensive, for example indicating how microbiome, enzyme activity and structure of the intestine differences could be due to other ecosystem factors besides temperature. A relation between different others water parameters, feed differences, and so could be an interesting point that will allow to a better understand how those ecosystems have been contributed to speciation.

SPECIFIC COMMENTS

  • A figure with a map pointing where both lakes are located in China will be helpful to understand how geographically far are.
  • In the “16s RNA sequencing and data analysis”, authors should indicate the reference of the primers used. Also, in this section title (L110) a “r” is missed , instead “rRNA” must be written.
  • In the section 3.2. authors mentioned that beta diversity is analysed, but this point is not mentioned in paragraph 136-145. An explanation of how this index is calculated must be incorporated to the text.
  • In Figure 1.A. the black lines and bars are difficult to see. Maybe, changing the colour to another clearer could help to see the lines better.
  • Table 1, could be incorporated to Figure 1, as another bar plot. In my opinion this may help to have a general overview about intestine structure and functionality. An alternative could be dividing these results in two figures: Figure 1 with the intestinal structure and the measures of the intestinal structure as bar plots, and Figure 2 with the digestive enzyme activities plot.
  • The statistical methodology should be improved, in order to explain how comparison of compositional microbiome is done. In the 3.3. section of the Results, it is said that a Wich’s t-test is done for intestinal microbiome comparison at phylum and genera comparisons and plotted in Figure 4. This analysis should be mentioned in the Material and Methods 2.3. section or in another separate epigraph in this section.

Author Response

Response to reviewer 2 comments

Comments and Suggestions for Authors

SUMMARY

The aim of this study is determining if there are differences in the intestinal microbiota, enzyme activity and structure of the intestine of two isolated populations of Culter alburnus from different geographical locations. And if those differences may be due from geographical isolation.

This manuscript contributes to the knowledge whether evolutive adaptation could significantly modify gut structure and enzymatic activity as well as microbiota of animals. This study shows that there are differences in microbiome composition and diversity and also in the physiology ad enzyme activity of gut. Finally, microbiome functionality analysis was done, in order to show if those microbiome differences are related with an adaptative response.

GENERAL CONCEPT COMMENTS

The word microbiota in the title should be changed to microbiome, which is more adequate to metagenomics analysis in which only 16S rRNA gene is used. Microbiota refers bacterial gut composition and microbiome to the OTUs composition.

Response: Thank you, revised as suggested!

In general, methodology procedures are well explicated. Only a few comments.

In the text is not mentioned how many individuals of each population (XK and DS) are sampled for this study. And if animals for each population are sampled from the same pound or from different pounds (L86). This point is important, because in all experiments is necessary have at least three biological duplicates of each treatment; in this case of each population.

Response: We collected ten individuals for each population (n = 10) and the ten biological replicates were used for 16s rRNA sequencing. The two populations (XK and DS) were sampled from two similar ponds in the Songjiang Aquatic Seed Farm (Shanghai, China).

Another study that could be interesting to done, maybe another time, is sampling wild individuals at different stations. This could give more information about gut parameters differences of each population due to their speciation to live in its environment. In the manuscript here presented,  in the results obtained of the offspring after being cultured during 6 years in controlled environment, is likely some species are loosed and maybe microbiome diversity has been modified, as well as intestine enzyme activity and structure. The sentence of L283-284 must be supported by some references.

Response: Thank you for your comments. We will conduct the related experiments in future research. We removed the sentences in L283-284 to avoid ambiguity.

This study has an interesting objective, that is dilucidated microbiome implication in fish adaptation to specific ecosystems, by comparing two different populations spatially differentiated. But, authors only apport temperature information about these two ecosystems. It may be attractive to know a bit more about other parameters that differentiate both lakes, and if those could explain the differences herein obtained in these two populations.

Response: Yes, there are many environmental factors except temperature that are different between the two lakes such as the lake area, flow rate, depth of water, plankton biomass that may also contribute to the differences of intestinal structure and intestinal microbiome between the two populations. We discussed them in the revised manuscript.

Finally, in my opinion, Discussion should be a bit more extensive, for example indicating how microbiome, enzyme activity and structure of the intestine differences could be due to other ecosystem factors besides temperature. A relation between different others water parameters, feed differences, and so could be an interesting point that will allow to a better understand how those ecosystems have been contributed to speciation.

Response: Thank you for the comments. We have discussed other environmental factors between the two lakes in the revised manuscript.

SPECIFIC COMMENTS

A figure with a map pointing where both lakes are located in China will be helpful to understand how geographically far are.

Response: We provided a new figure 1 in the revised manuscript that points the location of the two lakes.

In the “16s RNA sequencing and data analysis”, authors should indicate the reference of the primers used. Also, in this section title (L110) a “r” is missed , instead “rRNA” must be written.

Response: we have provided the reference for the primers used and ‘r’ was provided.

In the section 3.2. authors mentioned that beta diversity is analysed, but this point is not mentioned in paragraph 136-145. An explanation of how this index is calculated must be incorporated to the text.

Response: Thank you for the comments, we have provided detailed information for beta diversity analysis in the revised manuscript.

In Figure 1.A. the black lines and bars are difficult to see. Maybe, changing the colour to another clearer could help to see the lines better.

Response: Revised as suggested.

Table 1, could be incorporated to Figure 1, as another bar plot. In my opinion this may help to have a general overview about intestine structure and functionality. An alternative could be dividing these results in two figures: Figure 1 with the intestinal structure and the measures of the intestinal structure as bar plots, and Figure 2 with the digestive enzyme activities plot.

Response: Revised as suggested.

The statistical methodology should be improved, in order to explain how comparison of compositional microbiome is done. In the 3.3. section of the Results, it is said that a Wich’s t-test is done for intestinal microbiome comparison at phylum and genera comparisons and plotted in Figure 4. This analysis should be mentioned in the Material and Methods 2.3. section or in another separate epigraph in this section.

Response: We have provided related information in Material and Methods part in the revised manuscript.
